# Hydroclimatic Information Needs of Smallholder Farmers in the Lower Bengal Delta, Bangladesh

**Uthpal Kumar** [1,*], **Saskia Werners** [1], **Spyridon Paparrizos** [1], **Dilip Kumar Datta** [2] **and Fulco Ludwig** [1]

1   Water Systems and Global Change Group, Wageningen University and Research, P.O. Box 47,
    6700 AA Wageningen, The Netherlands; saskia.werners@wur.nl (S.W.); spyros.paparrizos@wur.nl (S.P.);
    fulco.ludwig@wur.nl (F.L.)
2   Environmental Science Discipline, Life Science School, Khulna University, Khulna 9208, Bangladesh;
    dkd_195709@yahoo.com
*   Correspondence: uthpal.kumar@wur.nl

**Abstract:** Hydroclimatic information services are vital for sustainable agricultural practices in deltas. They advance adaptation practices of farmers that lead to better economic benefit through increased yields, reduced production costs, and minimized crop damage. This research explores the hydroclimatic information needs of farmers by addressing (1) what kind of information is needed by the periurban delta farmers, and (2) whether information needs have any temporal dimension that changes with time following capacity building during coproduction of information services. Results reveal that the attributes of weather and water-related forecasts most affecting the farmers are rainfall, temperature, water, and soil salinity, along with extreme events such as cyclone and storm surges. The majority of the male farmers prefer one- to two-week lead-time forecasts for strategic and tactical decision-making; while female farmers prefer short-time forecasts with one-day to a week lead time that suggests the difference of purpose of the forecasts between male and female farmers. Contrarily, there is little preference for monthly, seasonal, and real-time forecasts. Information communication through a smartphone app is preferred mostly because of its easy accessibility and visualization. Farmers foresee that capacity building on acquiring hydroclimatic information is vital for agricultural decision-making. We conclude that a demand-driven coproduction of a hydroclimatic information service created through iterative interaction with and for farmers will enable the farmers to understand their information needs more explicitly.

**Keywords:** hydroclimatic information needs; farmers' decision-making; Lower Bengal Delta

## 1. Introduction

Hydroclimatic information services that involve timely production, translation, and provision of water, weather, and climate-related data, information, and knowledge for climate-sensitive societal decision-making are vital for agriculture and adaptation practices in deltas [1–5]. It help farmers to link their efforts to higher income, reduced inputs costs, and economic loss from climate risks and uncertainties [5–8]. The farming communities, however, in the Bengal Delta, currently stand on experience and traditional information base for agricultural practices and decision-making [9]. They do not have access to location- and time-specific information services in a meaningful way [9–11].

The Lower Bengal Delta—located in southwestern Bangladesh—is one of the most vulnerable deltas of the world due to climate change and sea-level rise [12]. According to Huq et al. [13], farming communities are highly vulnerable to different degrees of climate impacts in this delta. Smallholder farmers in the delta are highly dependent on rainfed agriculture. They face recurrent hydroclimatic

disasters such as cyclones, storm surges, tidal floods, waterlogging, and saline water invasion of the crop field [9,14]. These have immense impacts on the food production system and livelihood insecurity of the smallholders [13,15–17]. For example, the delta farmers were the worst victims of the Super Cyclone Sidr in 2007 [18–20], the devastating Cyclone Aila in 2009 [21,22], the Cyclone Bulbul 2019 [23], and the recent Super Cyclone Amphan in 2020 [24]. Besides extreme weather, the monsoon's onset is also critical for crop production in Bangladesh [25]. It affects the country's whole crop production system with a small temporal shift and variability [9]. Shahid [26], confirms a significant increase in annual average and premonsoon rainfall of Bangladesh. All these disaster events have had remarkable impacts on the agricultural production system. To adapt this hydroclimatic variability in the agricultural sector, boundary organizations such as the Department of Agricultural Extension (DAE) and local stations of the Bangladesh Meteorological Department (BMD) could play a vital role to increase forecast communication and uptake. However, they need the capacity to improve their forecast understanding and confidence to provide advisory services based on hydroclimatic information [27].

Currently, the existing hydroclimatic information services through governmental channels are not adequate to deal with frequent hydroclimatic disturbances in the delta. The services lack information quality due to traditional communication systems, short lead-time, and lack of user engagement [28]. Indeed, the forecast information is delivered as a one-way transfer of information to farmers that limit effective usage in agricultural decision-making [29]. These constraints related to credibility (i.e., perceived quality), salience (i.e., perceived relevance), and legitimacy (i.e., user interest) of the existing information services to influence farmers' decision-making [30]. Inwood and Dale [31], reported that the development of a digital decision support tool requires early and ongoing engagement and interactions with the targeted end-users. Farmers cannot also understand and respond to the available information services that often come late and lack of training [9,30,32,33]. A high degree of cultural belief, the experience of forecast inaccuracy, the reliance on tradition, and the local complexities in terms of weather patterns are some key factors that limit information uptake of farmers [9,11]. A need-based service coproduction is thus essential to deal with the recurrent hydroclimatic risks and disasters in the delta. Currently, the hydroclimatic information services platforms are mostly top-down, and have little or no uptake by end-users. Here, the platform means to create an improved information service in which users and producers interact to identify needs and capacities [4]. This suggests a need-based tailored information service for delta farmers [9].

To achieve the goal of climate services, understanding of farmers' information needs is an important aspect for tailoring hydroclimatic information services in a coproduction manner [2,33,34]. To do that, forecast lead-time is one of the key attributes for tailoring information. Forecast lead-time is classified as historical or past climate records, real-time or near real-time information, short-range weather forecasts of about one week, medium-range weather forecasts of about two weeks, and long-range weather forecasts and climate predictions such as monthly, seasonal and interannual time scales [35]. Forecasts less than a month, such as real-time information or near real-time, daily, weekly, and ten-day, are vital for agricultural decision-making. To date, many forecasting models struggle with predictivity below 10 days [36]. Gensini et al. [37], found that forecast skills are higher between two- versus three-week lead-time. Robertson et al. [38], developed calibrated probabilistic forecasts for northern India up to two weeks. They found appreciable skills for about one-week lead-time (days 3–9) with some skill at two weeks (days 10–16). In a seminal case study Gbangou et al. [39], notices that the European Centre for Medium-Range Weather Forecasts (ECMWF) System 4 seasonal climate forecast has a significant skill to predict seasonal onset variability at the local scale in Ghana, West Africa.

In the existing services, misalignment exists between information needs and information that is being provided to the intended end-users [40]. Literature suggests that a typical user survey is not enough to understand climate impacts and sensitivities in a region and to resolve mismatches between climate science products and user needs [41]. Here, user needs are defined as the hydrological

and meteorological parameters that are important to farmers for agricultural decision-making. This does not take place automatically without careful consideration, and it often requires iterative interaction and capacity building of the intermediaries and end-user farmers [9,42]. Available literature reveals that due to the disregarding of users' needs, information uptake is poor and less useable and useful to the targeted end-users [1,43,44]. However, assessing the information needs of farmers is not a simple task. The literature indicates that this is particularly difficult and time- and budget-demanding, and it includes relevant tasks for which people are either unfamiliar and/or have limited academic education and lack of capacity towards understanding and use such services for agricultural decision-making [45–47].

This research provides insight into the issue of 'understanding farmers' information needs' which is vital for better design information services for smallholders in agricultural decision-making [4]. Following the existing literature, we hypothesized that understanding information needs is not just a single step process. Information needs may change with time, such as the capacity building of the smallholder farmers. To test the hypothesis, iterative interaction and capacity building is an obligatory task to address information needs and better design information services for farmers. In this study, two research questions were addressed: (1) What kind of information is needed by the periurban delta farmers? (2) Do information needs have any temporal dimension that changes with time following capacity building during coproduction of information services with and for farmers in the delta? This research would help, besides farmers, researchers and policymakers to understand hydroclimatic information needs in a developing context where farmers and extension services have limited training and capacity on hydroclimatic information services for agricultural decision-making. This research would also help scientists, service providers, and developers for tailoring information for farmers in a coproduction manner.

## 2. Materials and Methods

### 2.1. Study Area

The study was conducted in three villages: *Jharbhanga*, *Sanchibunia*, and *Raingemari*, which are located in the Batiaghta *Upazila* (subdistrict) in periurban Khulna (Figure 1). The Khulna region represents the core of the Lower Bengal Delta. There were about 250 households in these three villages where 61% of the population are engaged in agriculture, 28% in service, and 9% in the industry sector. The region represents a low-lying coastal morphology that experiences frequent hydroclimatic hazards such as cyclones and storm surges, salinity invasions, and waterlogging [10]. The Rupsa-Bhairab-Pasur in the east and the Mayur river in the western boundary of the city play a key role in the agriculture-aquaculture farming systems of the area under investigation [48].

Farming around periurban Khulna comprises of three distinct crop seasons, the Rabi (November–March), Kharif-I (March–June), and Kharif-II (June–November). The region enjoys a subtropical warm and humid climate with four distinct seasons [26]. The average annual rainfall is 1752.3 mm and the mean annual temperature is 26.7 °C. The average monthly minimum and maximum temperatures are 21.9 to 31.3 °C, respectively. January is the coldest month with a mean minimum temperature of 12.9 °C and April is the warmest month of the year with a mean maximum temperature of 34.9 °C. About 80–90% of the rainfall takes place during the monsoon months of May to October [26,49]. The highest rainfall occurs in July and the lowest in December with monthly averages of 327.6 mm and 4.5 mm, respectively. High annual rainfall provides an excellent opportunity for agricultural practices in rain-fed conditions in the study area [50].

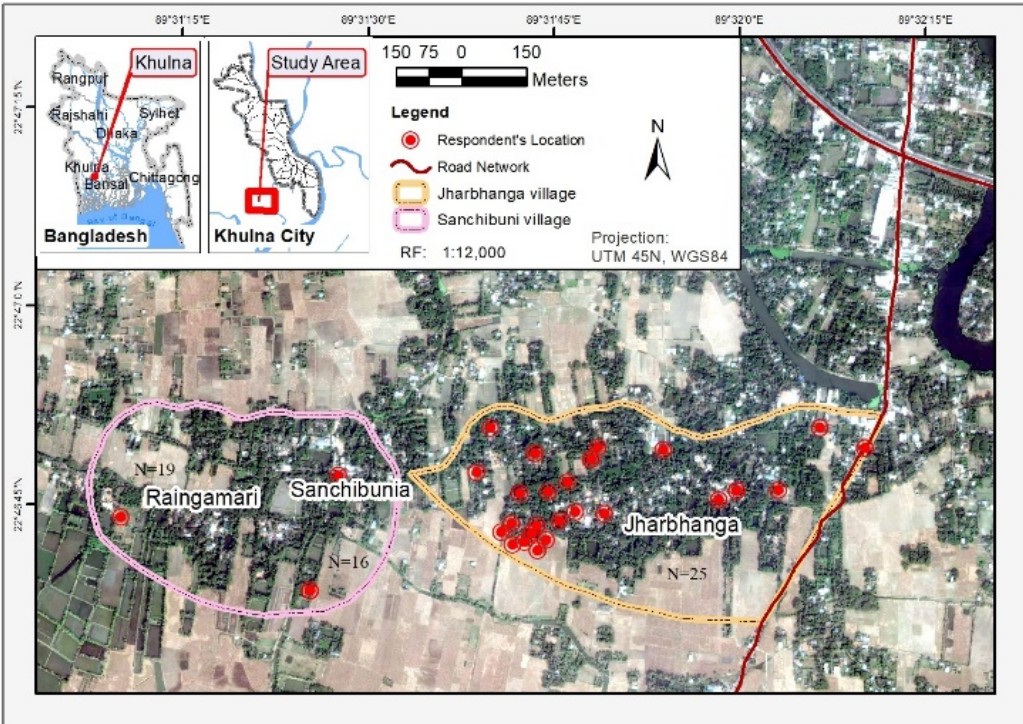

**Figure 1.** General location of the study area with point location of the sampled households or surveyed village locations in Batiaghta *Upazila* in periurban Khulna Bangladesh. At *Jharbhanga* village interviews (N = 25) were conducted at individual farmers' households. At *Sanchibunia* and *Raingamari* villages, interviews (N = 16 + 19 = 35) were conducted at three village points of the farmers' households.

## 2.2. Site Selection

A total of nine periurban villages was visited during reconnaissance visits. Five villages, *Badamtala, Mailmara, Jharbhanga, Sanchibunia,* and *Raingamari,* were visited in the Batiaghata subdistrict and four villages, *Mohishagunni, Sreefaltala, Domra,* and *Payara* were visited in the Rupsa subdistricts. Farmers were interviewed randomly during reconnaissance visits to explore local cultivation practices and to gain prior knowledge of the prevailing hydroclimatic vulnerability. We observed a similar cultivation pattern at all periurban villages surveyed that involves two major rice crops (*T-aman* and *Boro*) with varieties of vegetable cultivation practices round the year at homesteads and agriculture-aquaculture farming systems. Subsequently, three periurban sites *Jharbhanga, Sanchibunia,* and *Raingamari* were selected at the Batiaghata subdistrict for further interactions and capacity building of farmers and data collection for this study, which are depicted in Figure 1. The selected villages are easily accessible from the city of Khulna due to proximity and they additionally present typical periurban physiognomies for the general area, having both rural and urban landscapes, socioeconomic linkages, and livelihood interdependency.

## 2.3. Data Collection

The primary data were collected through baseline and endline assessments conducted from February 2018 to April 2019. The schematic representation of the data framework is depicted in Figure 2; Figure 3. The baseline assessment includes reconnaissance field visits and meetings with farmers and semi-structured personal interviews with 60 periurban farmers selected for this study (Supplementary Materials A). These activities help us to understand the information needs of selected farmers' groups at baseline conditions. The endline assessment includes a second round of personal interviews with the same questionnaire and farmers' groups, followed by farmers' engagement, training, and weekly interaction meetings. Results from the baseline and endline assessment were compared to

discourse farmers' needs and to assess how that information needs change with interactions and capacity building of farmers. The farmers that were selected for this study had experience in hydroclimatic information services using a smartphone and their application in agricultural decision-making. Thus, the farmers' capacity building training was conducted at the study villages through farmer field schools (FFS) with the help of the local extension office (DAE). FFS is a group-based agricultural extension approach of DAE.

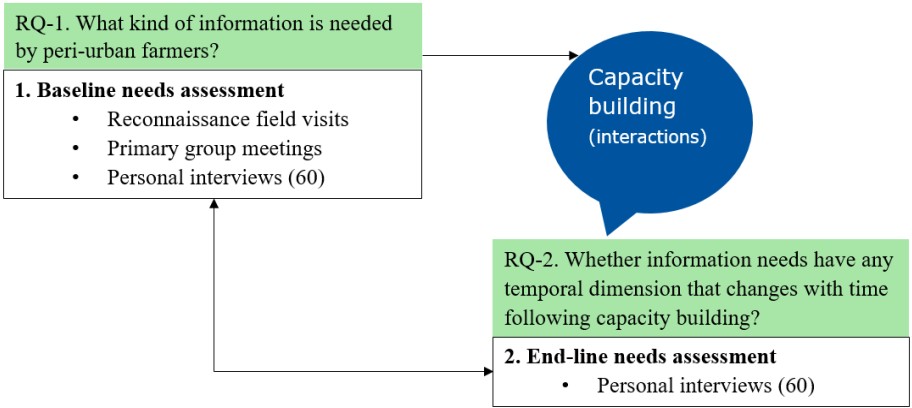

**Figure 2.** Primary data collection framework including the steps and participatory tools that were employed in the current study.

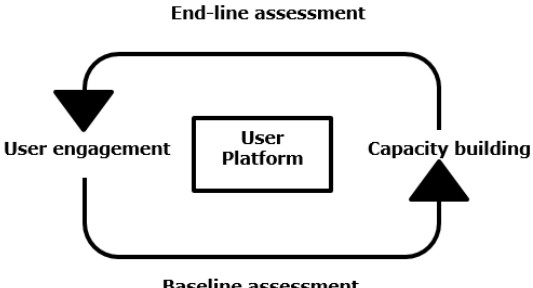

**Figure 3.** A five-step farmer engagement process for understanding the information needs of smallholder farmers in the Lower Bengal Delta.

We engaged two farmers' groups through FFS during three crop seasons (Kharif-I: March to July; Kharif-II: July to November; Rabi/Boro: November to March) and provided Meteoblue [51]) forecasts for seven and 14 days, and seasonal (three-month) weather forecasts as well as training on their interpretation and advisory services with the help of the local agricultural extension office. An example of the provided Meteoblue forecast is given in (Supplementary Materials B). Three kinds of face-to-face interaction were provided through the FFS: (1) general discussion and comments on the weekly provided forecast information (seven-day, 14-day and three-month forecasts); (2) group learning through elaboration and discussion on the forecasts and their interpretation; and (3) agricultural advisory and decision-making based on the forecasts in a participatory way together with the experts from local agricultural extension department (Supplementary Materials C). Relevant secondary data were retrieved through desk research from relevant research articles, databases of the Bangladesh Bureau of Statistics (BBS), and unpublished database and reports of the local extension office.

A five-step user engagement process helped us to understand the information needs of smallholder farmers and to collect primary data through an interactive process for this research (Figure 3). The user engagement and capacity building are two main iterative components that are activated under users' platforms such as FFS [6,9]. Finally, a baseline and an endline assessment were conducted to compare information needs for better design information services with and for the farmers.

### 2.3.1. Baseline Needs Assessment

Three primary meetings were conducted at *Jharbhanga, Sanchibunia*, and *Raingamari* villages. The meetings were conducted with the listed farmers' groups of the local extension office. There were 25–30 households in each farmers' group, which represented 50–60 farmers, considering husbands and wives as group members. For the primary meetings, we invited a single participant (either male (husband) or female (wife)) from the farmers' households. We engaged participants for about 2 h for mapping: (i) major crops, (ii) key decision points, and (iii) agricultural information sources in a flipchart paper. We discussed what they do to confront hydroclimatic hazards and how they currently access hydroclimatic information services for agricultural decision-making. The major hydroclimatic hazards indicated by farmers were untimely heavy rainfall, waterlogging, cyclones, thunderstorm, hailstorm, temperature stresses (drought and cold stress), heavy fog, salinity invasion, etc. Finally, we discussed information needs, forecasts availability, lead-time, and their future interest in information services for agricultural practices and decision-making. At this instant, the lead-time denotes forecast information in sufficient advance time scales so that farmers can take action for saving crops and other livelihood assets from hydroclimatic risks and variability. In these meetings, we got an idea about hydroclimatic risks and climate-sensitive key decisions of farmers. We also conducted a primary meeting with the district and subdistrict extension officers of the DAE to discuss and triangulate the hydroclimatic information needs by farmers. The extension officers (N = 6) commented on information needs for farmers, current availability, and required lead-time. After the primary meetings, a total of 60 farmers were interviewed using a semistructured questionnaire for a detailed understanding of their needs for managing hydroclimatic risks and uncertainties. Among them, N = 25 of farmers were from *Jharbhanga* village, N = 16 farmers were from *Sanchibunia* village and N = 19 farmers were from *Raingamari* village. Figure 1 shows farmers' locations in the study villages. The questionnaire focused on the weather- and water-related information needs, forecasts' lead-time, the preferred method of communication, and the format of the needed information (Supplementary Materials A). The respondents were selected randomly from the listed farmers' groups of DAE in the selected villages. At least 30% of participants we considered were female.

### 2.3.2. Endline Needs Assessment

After capacity building, training and frequent interaction with farmers, we noticed changes in hydroclimatic information needs, lead-time, and parameters of interest such as a thunderstorm, hailstorm, water and soil salinity, cold stress, fog, etc. Lead-time implies advance forecast information in sufficient advance time scales such as (sub-) daily, weekly, monthly, seasonal, etc. for taking climate-sensitive decision-making. Following changes in farmers' information needs, we interviewed the same experimental farmers' groups for the second round for their endline needs assessment. All these farmers also participated in the weekly interaction and training we had initiated through FFS for the coproduction of hydroclimatic information services with and for periurban delta farmers.

### 2.4. Data Analysis

The collected qualitative data such as field observation notes, group meetings notes, and farmers' comments were organized, coded, summarized in Word files, and Excel datasheets for further analysis and interpretation. The primary quantitative data that were collected through interviews were then coded and analyzed using the SPSS software (Version 23) for the preparation of a results summary in tabular format. The graphs were generated from the quantitative results using python 2.7 software. The study area point data were collected using Google Earth and finally, the map was prepared using ArcGIS software.

## 3. Results

### 3.1. Demographic Profile of Farmers

The demographic details of the farmers are presented in Table 1. Female farmers were mainly involved in homestead agricultural activities that include cultivation of vegetable, livestock rearing, compost preparation, crop processing, storage, and other supportive activities for field crops. In contrast, the male farmers were involved in field crop production, day-to-day farm management, and tactical decision-making. The majority of the farmers were between 26–40 (37%) and 41–60 (35%) years old. A total of 22% of farmers were above 60 years old and a few (7%) were young farmers between 18–25 years old. Education status shows that a total of 35% of farmers were educated at the primary level, 27% were educated at the secondary level and 20% were educated at the higher secondary level. A total of 18% of farmers had no formal academic education. The average household size of the farmers was five persons. In detail, 63% of farmers had a family size of three to six people, 17% had a family size of one to two people and 20% had a household size above six people. In the study sites, most farmers had more than 10 years of experience in farming activities. A total of 33% of farmers had farming experience between one and 10 years. Among interviewed farmers, 65% of farmers have leased lands for agricultural practices. However, 77% of the total farmers had personal land ownership and 33% of farmers had no land ownership for agricultural activities. Monthly income range indicated that about half of the farmers could earn five thousand Bangladesh Taka (BDT) or less which is less than two US dollars per day. A total of 23% of farmers were fully dependent on agricultural incomes and the majority of them (77%) were involved in off-farm incomes in the cities and peripheral areas. The average farm size of smallholder farmers was 182 decimals with a minimum and maximum farm size of 6 and 675 decimals, which includes personal and lease lands.

**Table 1.** Descriptive statistics of the respondents (Source: Baseline survey).

| Variable | N = 60 | % | Variable (N = 60) | N = 60 | % |
|---|---|---|---|---|---|
| **Gender** | | | **Farm Experience** | | |
| Male | 38 | 63 | 1–10 years | 20 | 33 |
| Female | 22 | 37 | 11–20 years | 18 | 30 |
| **Age** | | | 21–30 years | 7 | 12 |
| 18–25 | 4 | 7 | Above 30 years | 15 | 25 |
| 26–40 | 22 | 37 | **Land ownership** | | |
| 41–60 | 21 | 35 | Yes | 40 | 67 |
| Above 60 | 13 | 22 | No | 20 | 33 |
| **Education** | | | **Monthly income (BDT)** | | |
| Primary | 21 | 35 | 1–5000 | 31 | 52 |
| Secondary | 16 | 27 | 5001–10,000 | 15 | 25 |
| HSC | 12 | 20 | 10,001–20,000 | 14 | 23 |
| Illiterate | 11 | 18 | **Off-farm income** | | |
| **Family size** | | | Yes | 46 | 77 |
| 1–2 | 10 | 17 | No | 14 | 23 |
| 3–6 | 38 | 63 | **Farm size (decimal)** | | |
| Above 6 | 12 | 20 | Average | 182 | |
| | | | Minimum | 6 | |
| | | | Maximum | 675 | |

### 3.2. Hydroclimatic Challenges and Information Needs of Farmers

Information gathered from primary meetings is summarized in Table 2. The major products during Kharif-I are short-duration vegetables and oilseeds, while rice is the main product for the Kharif-II (*T-aman* rice) and Rabi (*Boro* rice) seasons. Farmers face different hydroclimatic challenges during the three crop seasons. The Nor'wester (thunderstorms)—locally named *Kalboishakhi*—during Kharif-I is characterized by sudden thunderstorms, hailstorms, and cyclones with heavy rainfall. Additionally, drought and scarcity of irrigation water is also a common characteristic of Kharif-I crop season in the entire delta. Water in the rivers and canals is highly saline during the Kharif-I as it was reported by the local farmers. On the other hand, the Kharif-II season is characterized by heavy rainfall, waterlogging, storm surges, and cyclones. The majority of the farmers reported that they cannot go to the crop fields regularly during the Kharif-II season due to heavy rain and storms. The Rabi crop season remains cold and dry with frequent cold spells, drought, and intense fog. Farmers indicated that winter rainfall and cyclone events during the Rabi season often damage mature rice fields and that have been more frequent in the study area in the last few years.

**Table 2.** Summary of the season-specific hydroclimatic information needs of smallholder farmers in the Lower Bengal Delta.

| Crop Seasons | Major Crops | Hydroclimatic Challenges | Information Needs | Forecast Lead-Time |
|---|---|---|---|---|
| Kharif-I (Summer) (Mid-March to Mid-June) | Vegetables and oil seeds | Nor'wester or Kalboishakhi, hailstorm, drought, water scarcity, etc. | Rainfall, thunderstorm, and cyclone | Seasonal; monthly; 1–2 weeks; 1–3 days |
| Kharif-II (Monsoon) (Mid-June to Mid-November) | Taman paddy and summer vegetables | Heavy rain or less rain, drought, waterlogging, storm surge, and cyclones | Rainfall, dry days, temperature, thunderstorm, and cyclone | |
| Rabi (Winter) (Mid-November to Mid-March) | Boro paddy and winter vegetables | Cold spells, storm surge, and cyclones, winter rain, fog, drought, etc. | Rainfall, cold spells, sunshine duration, thunderstorm, and cyclone | |

Local farmers reported that an indication of weather information, such as rainfall intensity and precise timing and cyclone and thunderstorm signals, one to two weeks in advance would reduce their unexpected crop damages through taking climate-sensitive decisions. In addition, an indication of monthly to seasonal weather forecasts would help farmers to make strategic and more sophisticated decisions such as crop selection, cultivation decision, land allocation, and choice of crop variety. For example, sesame is commonly cultivated during Kharif-I in the study area. Sesame is a highly profitable and short duration crop to farmers. However, if farmers have indications about heavy rainfall occurrences in Kharif-I, they would not go for large-scale sesame production particularly in low lands. Sesame fields are sensitive to waterlogging conditions. On the other hand, if farmers perceived a very heavy rainfall during Kharif-II in advance they would prefer to grow local varies of T-aman rice instead of hybrid rice in the low lands. The stem height of hybrid rice is low, thus expected production might be affected due to the waterlogging problem. Overall, in the primary meetings, farmers reported lead-time hydroclimatic forecast information on rainfall, temperature, thunderstorms, cyclones, cold spells, and sunny or dry weather days. About 90% of the farmers revealed that one to two weeks of advanced forecast would reduce their crop damages by 60–80% in the harvest period. However, about 60% of farmers also indicated that thunderstorm and rainfall forecasts one to three days in advance would also reduce major hydroclimatic risks and damages through taking tactical

decisions such as hazard preparedness, collection of harvested crops, repair of farmers' households, repair of farmhouses, and keeping farm assets and livestock at a safe place. Farmers said that they do not allow livestock in the field during a thunderstorm and rainy weather conditions. They could also manage fodder for livestock if they perceived thunderstorms, cyclones, and rainy weather conditions for consecutive days.

*3.3. How Did Information Needs Change between Baseline and Endline Interviews?*

### 3.3.1. Weather-Related Information Needs

The weather information needs of farmers shifted from the baseline assessment (Figure 4) except for that of information on rainfall (such as monsoon onset, amount, duration, etc.). During endline assessment, about half of the farmers indicated that forecasts on high-temperature and intense rainfall are two important parameters of interest to predict drought and waterlogging situations in advance. This would help farmers to save crops from drought and waterlogging conditions by taking advanced measures such as subsistence irrigation and excavation of drainage systems in crop fields. Paddy farmers said that sometimes irrigation by brackish water is inevitable during a prolonged drought period. In contrast, they use a pump to get rid of waterlogging, especially during the ripening stage. The majority of the farmers during endline assessment indicated that forecasts of hailstorms, cyclones, and storm surges and fog are vital for agricultural decision-making. This means that information needs may change over time due to the capacity building of farmers. Thus, to tailor information services for farmers, capacity building is required to understand the detailed information needs of the smallholder farmers.

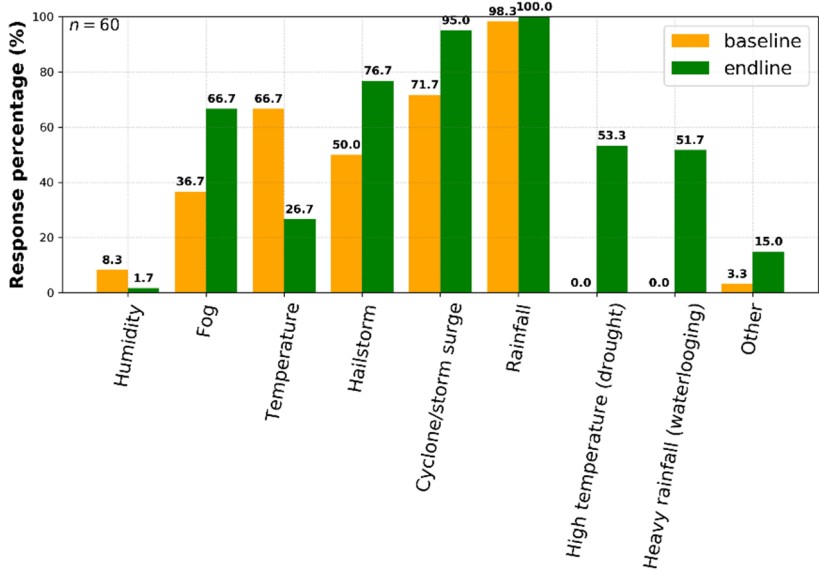

**Figure 4.** Baseline versus endline weather information needs of smallholder farmers in the Lower Bengal Delta around Khulna, Bangladesh.

### 3.3.2. Water-Related Information Needs

The majority of the farmers during the baseline needs assessment indicated that they require forecasts on the flood, water and soil salinity, and river discharge (Figure 5). This information could assist them in selecting crops based on land conditions. Such forecasts help them in selecting local varieties that are more resilient to flooding conditions, while, during endline interviews, only 2% of the farmers reported flood forecasts as an information need and none of the farmers reported river discharge as an information need. The farmers realized that, since their paddy fields are located in a poldered (embanked) entity, they are not susceptible to recurrent and severe floods and are well protected

from high river discharge during the monsoon. Indeed, farmers in these villages face short-term waterlogging and drainage congestion during intense rainfall. They added that the waterlogging is mainly due to encroachment of the natural drains and canals by local elites, who often acquire these natural drains and canals for aquaculture through the governmental leasing process. Frequent interaction and training may improve farmers' capacity to address these aforementioned local issues and information needs. This concludes that limited interaction with farmers is not sufficient to identify the appropriate information needs for tailoring hydroclimatic information services. Endline assessment also confirms that water and soil salinity is the most important parameter of interest to farmers for agricultural decision-making at the local context to deal with increased hydroclimatic vulnerability.

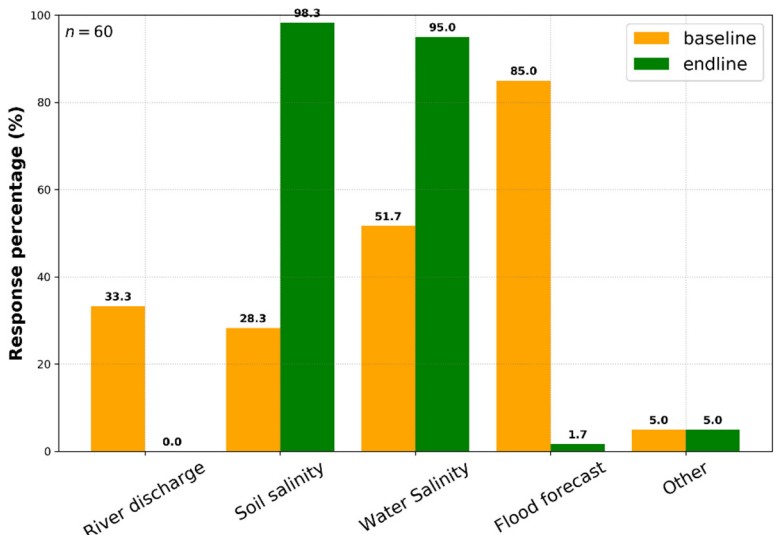

**Figure 5.** Baseline versus endline water information needs of periurban farmers in the Lower Bengal Delta.

### 3.3.3. Lead-Time Information Needs

About half (48%) of the farmers were interested in receiving two-week to one-month forecasts in advance for agricultural decision-making during the baseline needs assessment (Figure 6). However, a total of 38% of farmers were interested in receiving weekly forecasts and 42% were interested in real-time forecasts. A few of them (18%) opted for seasonal (three-month) and two to three day advanced forecasts (15%). However, during endline assessment, farmers indicated that they mainly take agricultural decisions for one to two weeks at current practices. Thus, they were less intended for a monthly to seasonal scale planning culture for agricultural decision-making. Besides, the traditional rice farmers said that they do not take decisions for more than one to two weeks in advance. They reported monthly and seasonal scale forecast accuracy was not useful in their current agricultural decision-making practices. We also observed that forecast accuracy significantly differs between the one-week and two-week timescale. Very few farmers (3.3%) reported that the monthly to seasonal forecasts were useful for agricultural decision-making. Figure 5 shows that needs for one to two weeks of forecast were expressed by the majority of the farmers in the study area. This concludes that farmers require less but more specific information than what they have expressed during the baseline assessment. They also reveal that monthly to seasonal scale information quality is not good enough for precise decision-making. A short lead-time forecast (one week or less) is more appropriate and applied for precise agricultural decision-making.

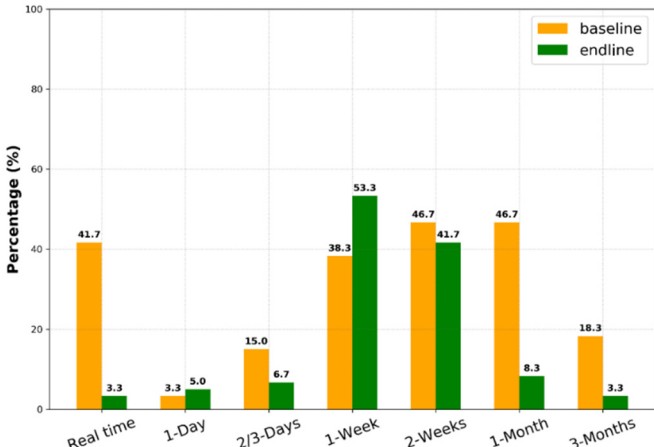

**Figure 6.** Baseline versus endline lead-time of hydroclimatic information needs of smallholder farmers in periurban areas of the Lower Bengal Delta.

### 3.3.4. Choice of the Communication Platform

Farmers do not prefer traditional information platforms such as radio and television for agricultural decision-making (Figure 7). Currently, ICT-led platforms such as smartphones and social media are more preferable and accessible for weather and climate-related information services. In the face-to-face meetings during the FFS, farmers reported that forecasts (i.e., Meteoblue; see Supplementary Materials B) through a smartphone do not require extra time to access information. In the baseline and endline comparison, results revealed that after capacity training and interactions, 81% of farmers were interested to receive weather information services through a smartphone. However, before training, the majority of the farmers (85%) stated that they lack ICT skills to receive information through a smartphone. Hence, farmers' interaction and training could overcome the existing usage barriers concerning access to ICT tools such as smartphones and help farmers to efficiently uptake hydroclimatic information for agricultural decision-making.

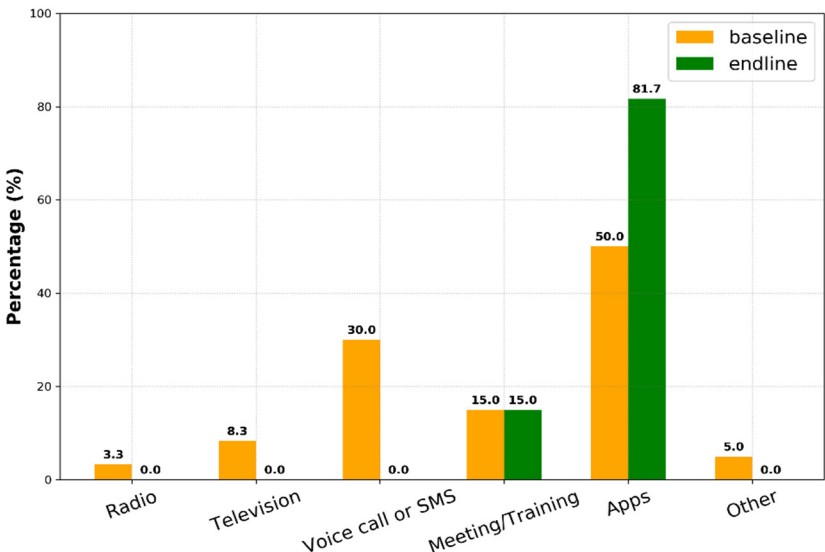

**Figure 7.** Farmers preferred communication platforms for receiving hydroclimatic information service in periurban areas of the Lower Bengal Delta. Here 'other' includes in-person communication with individual farmers or lead-farmers in a village for delivering hydroclimatic information services by the field extension officers.

### 3.3.5. Format of the Information

The results indicate that farmers prefer more visualization for receiving forecasts such as photographs or diagram based rather than a written text format (Figure 8). They opined that the visual format is easy to understand by looking at the symbols for different weather phenomena such as rain, thunderstorm, sunshine, etc. Initially, the farmers raised some concerns about the use of visual images to gain a clear understanding of activities during the succeeding weeks. However, through iterative interaction, farmers showed a clear interest in a diagrams based forecast than in the text-based written information (See Supplementary Materials B). This was also more useful for people with little or no literacy, as reported by the farmers during FFS and endline interviews. The bar-plots that depicted probability percentages (%) in the visual diagrams were more acceptable than the line plots, to make them understand the severity and accuracy of the forecasts and the predicted events.

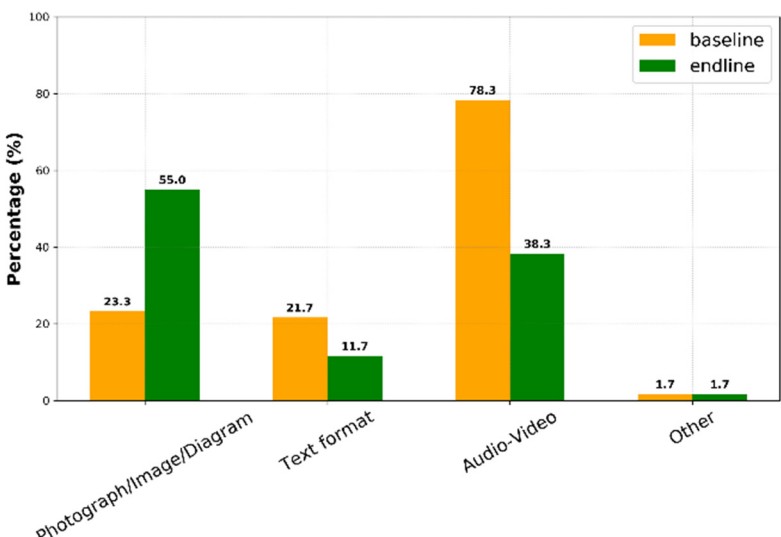

**Figure 8.** Farmers' preferred formats for receiving hydroclimatic information service in the Lower Bengal Delta.

## 4. Discussion

### 4.1. What Kind of Information is Needed by Farmers?

Farmers face different kinds of hydroclimatic challenges in the Lower Bengal Delta (Table 1). The results show that information needs have four key dimensions: (1) water or hydrological information, (2) meteorological information, (3) soil quality information, and (4) emergency weather forecast. During interviews and focus group meetings, farmers reported that information forecasts on these four key dimensions would help them in strategic and tactical decision-making. The local extension officers, however, indicated that together with the weather- and climate-related information, farmers also need advisory services on technology and agronomic variables such as modern cultivation methods and diversified crop varieties for precise agricultural decision-making based on local hydroclimatic conditions [52]. All this information is vital and interconnected for agricultural decision-making. Stone and Meinke [53] also observed that farmers need advisory services on seasonal variability to adjust farming practices that best suit the upcoming season and take advantage of weather forecast information [52]. The majority of farmers are currently facing challenges while taking agricultural decisions due to increased hydroclimatic variability [11]. For example, some farmers at *Jharbhanga*, *Sanchibunia*, and *Raingamari* villages stated that "the current weather does not follow any traditional rules, and we often made wrong decisions based on our traditional understanding".

In the current study, the hydroclimatic information needs of farmers were compared between baseline and endline assessment. The results indicate that farmers expressed a more general outlook during the baseline information needs assessment. The key information needs were rainfall, cyclones and storm surges, hailstorms, fog, temperature, and humidity. However, during the endline needs assessment, farmers were more specific to express their information needs, required timescales, and preferred platforms. This might impact capacity building through training as well as farmers' engagement and frequent interactions with forecast services. For example, during the endline assessment, farmers reported that forecasts on extreme temperature and heavy rainfall were more important for managing drought and waterlogging risks. The majority of farmers indicated that emergency forecasts such as cyclone formation and storm surges are vital for their tactical decision-making.

Boekel [54], found out that there is a huge gap between information needs and available information for sesame farmers in the study area. Bernardi [55] reported similar results with farmers of eastern Australia where the available forecast does not meet the farmers' needs. Interestingly, Boekel [53], did not address any hydroclimatic information needs of the sesame farmers in the Lower Bengal Delta. However, during the current study, all farmers and extension officers reported that sesame is the most affected crop in this delta area due to weather variability such as a sudden heavy rainfall during the Kharif-I (pre-monsoon). Farmers and extension officers reported that sesame is one of the most popular crops in the entire delta due to high price and short crop cycle. However, currently, sesame cultivation has been reduced due to an increased trend of heavy rainfall in the delta during the Kharif-I season [10,56].

Information lead-time may vary among users such as farmers, agency owners, and policymakers. For strategic and policy-oriented decision-making, the forecast lead-time may include a short-time-scale forecast (weekly to bi-weekly), the climate prediction in a medium-time scale (monthly to seasonal, and decadal-scale), and the climate projection in a long time-scale (10–30–50 years) [54,57]. In this research, forecast lead-time includes weekly, bi-weekly, and seasonal lead-time for farm scales decision-making. To assess that we used an open-ended response for providing more freedom to the interviewee farmers regarding their needed time-scale for hydroclimatic information services. However, we found a shift of forecast lead-time between baseline and endline assessment. The forecast lead-time was more inclined towards weekly and bi-weekly scales during the endline needs assessment. Smallholder farmers reported that they were well experienced in traditional cultivation seasons and cropping practices. Thus, they mostly need a short timescale information service to help them in tactical and operational farm management decisions based on local hydroclimatic variability. Only the farmers with academic education and a good farming knowledge (~12%) reported that monthly or seasonal scale information would help them for taking more strategic decisions such as crop selection, land allocation (high and low land and amount to be cultivated), seasonal water availability (quality, quantity, and timing) and input collection in advance for the upcoming season. The female farmers, however, requested one to three days' forecast information for household and farming activities such as vegetable production, processing crops, and firewood, dung-stick preparation (for fuel), fodder collection, and management of livestock and farm assets.

Similar studies, such as Nyadzi et al. [5], found that farmers in northern Ghana preferred information on a monthly scale before the season for strategic decision-making. However, in the Lower Bengal Delta, most farmers prefer one to two weeks of lead-time for strategic as well as tactical decision-making. A few farmers with academic education and knowledge on cultivation requested monthly to seasonal forecasts for strategic decision-making. Farmers mentioned that cyclone and storm surge forecasts are crucial, especially during the paddy harvesting period. They reported that a good forecast during the paddy harvesting period could reduce crop damages up to 60–80% that also fundamentally liked the food security issue of the smallholder farmers of the delta. The forecast lead-time needs may also be different based on field location, farm size, ease of transportation, and expected crop yields added by the respondent farmers during interviews. For example, if a farmer's

field is located at a remote distance and the farm size is large enough, then more advanced information is needed compared to a small paddy farm located close to the farmers' households. On the other hand, if farmers do not expect satisfactory production then they are less interested to invest in labor and transportation costs for crop harvest and collection, and they require more advanced forecasts for harvest by themselves. The female farmers, in contrast, reported that a weekly lead-time forecast is enough for their households' level of farming and livestock management activities. Carr, Fleming and Kalala [47] also reveal that even at the village level, the women have different information needs and forecast lead-time. The gender-sensitive information needs thus should be considered to improve uptake (access and use) of information services for agricultural decision-making [30,42]. Besides gender, Carr et al. [8], and Roncoli et al. [58], revealed that the sociocultural factors also influence farmers' engagement process and uptake of hydroclimatic information and they should be carefully addressed during information provision and service coproduction.

In the primary meetings, farmers did not discourse about the choice of communication platform and format for the forecast information. However, during baseline and endline assessment, we addressed these two important design principles with the local farmers that indicated a significant change between baseline and endline assessment. After capacity building, mobile-app (82%) and diagram-based (55%) forecasts were mostly preferred by the majority of farmers (see Figures 7 and 8). Kumar et al. [9] found that about 54% of farm households already have access to smartphones in the study area, and the majority of them were connected to a social media app such as Facebook. Thus, a smartphone app such as social media could play a vital role in providing forecasts and advisory services to farmers translating information into the local language. After capacity building, experimental farmers also requested to continue sharing the Meteoblue forecast diagrams (seven days, 14 days, and three months) through creating a messenger group. In addition, farmers said that they would share these forecasts with their family, peers, and relatives involved in agricultural activities. In this direction, Inwood and Dale [31], also found a need for mobile applications as a digital decision support tool for emphasizing knowledge exchange rather than just some information delivery. However, in sub-Saharan Africa, Feleke [59], found that more than half of the farmers still depend on the radio as the preferred communication platform. A few model farmers knew about the technology and its uses in sub-Saharan Africa. Farmers in Zambia also requested weekly to seasonal forecasts through community radio [60]. This indicates that the choice of platform and information lead-time may vary from place to place during the coproduction of location-specific hydroclimatic information services [52]. In the study area, ICT-led platform such as mobile app is most preferred but traditional platforms such as FFS, radio, and television may also have a wider application based on community preferences, availability of technology, socioeconomics, and cultural perspectives.

*4.2. Do Information Needs Change over Time and Capacity Building?*

4.2.1. Knowledge Improvement of Farmers

The results revealed that capacity building has increased the knowledge of farmers about hydroclimatic information (Figure 9). During the baseline needs assessment, the majority of farmers' knowledge of hydroclimatic information was poor (*n* = 45%) to moderate (~48%). However, during the endline assessment, farmers (~80%) reported an increased knowledge level due to interactions and capacity building. Among them, about 68% of farmers claim that they currently have a good knowledge and 10% of farmers claim an excellent knowledge of hydroclimatic information. Moreover, farmers also reported a wide range of limitations towards the uptake of the available hydroclimatic information during the baseline assessment (Figure 10). However, following interaction and capacity building, farmers only reported the lack of smartphones and the lack of ICT knowledge as their key limitations to uptake information. Other limitations such as economic reasons, incompatible design, internet unavailability, etc. weren't reported by farmers during the endline needs assessment. Interactions and capacity building of farmers improved their knowledge base as well as their understanding, which

helped them to express needs more precisely based on local hydroclimatic vulnerabilities. Sultan et al. [32], found out that lack of training and capacity building is an important barrier apart from the stakeholder engagement for better uptake information services by the end-users.

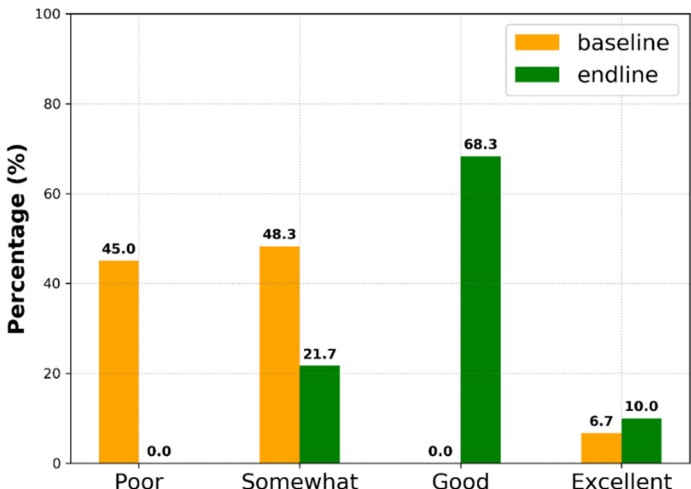

**Figure 9.** The baseline and endline comparison of farmers' knowledge of hydroclimatic information in the study area.

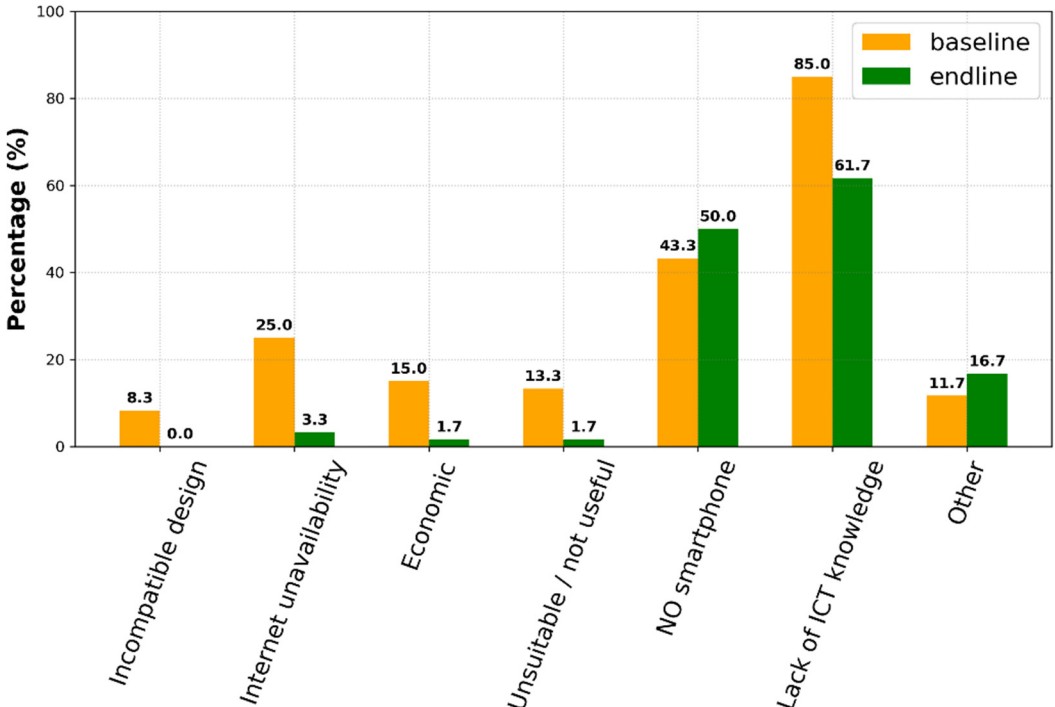

**Figure 10.** Limitations of uptake available hydroclimatic information by periurban farmers in the Lower Bengal Delta, Khulna, Bangladesh.

#### 4.2.2. Farmers' Engagement Process

There is no ideal approach for user engagement [45]. This study engaged farmers and agricultural extension services. Lack of engagement with relevant end-users limits the effective uptake of information services [1,55]. Similar studies found a critical role of iterative interaction between end-users and service producers for understanding information needs [45,46,61]. They also revealed that the lack of user understanding and capacity limit the uptake of forecasts information in decision-making.

In the current study, a participatory engagement process was followed where farmers and local extension officers were actively involved in knowledge coproduction [61]. The results showed that user engagement not only builds trust and a positive relationship between the information provider and users but also improves community confidence to deal with the frequent hydroclimatic hazards and to better respond to the provisioned forecasts and provide feedback on the overall quality of the information services. Initial engagement of farmers through the farmers' field school (FFS) provided an excellent opportunity for service coproduction in a participatory way where users discussed their specific needs, accessibility, relevance, and particular usage [6]. Results revealed that farmers often had difficulty expressing their needs, particularly as they were not familiar with hydroclimatic information services and have limited or no academic literacy [46,59]. However, the FFS platform helped towards addressing this issue through weekly interaction and participatory training with the help of the local extension office.

## 5. Conclusions

This study confirmed that information needs shifted over time as farmers gained better understanding and experience on the use of location-specific hydroclimatic information services. A better understanding of information needs is thus vital, requiring participatory interaction and capacity building. It also helps smallholder farmers to better understand forecast information for climate-sensitive decisions following capacity building and frequent interaction during the coproduction of information services. A series of participatory interactions enabled farmers adequately to express their specific information needs. We thus conclude that understanding of needs is not a single-step process. It should consider a baseline and endline assessment with capacity building training, considering all crop seasons, local agricultural practices, socioeconomics, culture, and technological perspectives of the users. Tools such as Participatory Integrated Climate Service for Agriculture (PICSA) can be useful for step by step interaction and capacity building training for farmers [62].

Secondly, why and how did the needs change and by what processes? This study concludes that change in needs resulted when farmers had a better understanding through a series of participatory interactions and the provision of information services help through a learning process over time. We followed a coproduction approach that followed with stepwise farmer engagement processes (see Figure 3). The engagement process and steps may vary, based on the capacity and local contexts of the target audience. For example, financial organizations and policymakers require information on a regional and long-term perspective such as climate predictions (monthly to seasonal) and projections on a (multi-) decadal scale. They may not need participatory interactions with baseline and endline assessment. This study was performed with a limited number of farmers in a coastal periurban context in the lower Bengal Delta. Thus, the hydroclimatic challenges and information needs might be different in other regions and or even in the same region as the users can access and uptake weather and climate information services more frequently and with higher local precision. Cultivation practices, seasonal variability, technology, and other socioeconomic dimensions also need special attention to understand the needs and to better design information services to influence farmers' decision-making. Finally, we conclude that a primary need assessment is not adequate for developing hydroclimatic information services when farmers do not have a proper understanding of what it is delivered at the community level, and what they need. An iterative approach thus can provide the best outcome for understanding farmers' needs with capacity building and coproduction information services with and for farmers.

**Supplementary Materials:** The following are available online at http://www.mdpi.com/2073-4433/11/9/1009/s1, Supplementary Material A: Farmers' baseline/endline needs assessment questionnaire, Supplementary Material B: Waterapps Climate School Khulna, Bangladesh—Meteoblue ensemble forecasts, Supplementary Material C: Waterapps Climate School—Farmers Field School (FFS) Protocol Expert Interview Form.

**Author Contributions:** U.K. conducted this study supervised by S.W., F.L., D.K.D., and S.P. The supervisors contributed substantially in the study design, editing and commenting on the draft article for several rounds, and visited field sites during data acquisition of this study. All authors have read and agreed to the published version of the manuscript.

**Funding:** This research is an output of the "Waterapps—Water Information Services for Periurban Agriculture" project funded by the Netherlands Organization of Scientific Research (NWO) under its Urbanising Deltas of the World (UDW) program, grant number W 07.69.204.

**Acknowledgments:** We are highly indebted to funders, partners, project coordinator Erik van Slobbe and for their worthwhile contributions. We acknowledge DAE officials and farmers for their valuable time and response during field visits, interviews, and FFS sessions. We also thank EsrAz Ul Zannat for his kind assistance in preparing the location map for this study.

**Conflicts of Interest:** The authors declare no conflict of interest.

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
