# Peer review of "Hydroclimatic Information Needs of Smallholder Farmers in the Lower Bengal Delta, Bangladesh"

_atmosphere, doi:10.3390/atmos11091009_

Round 1
Reviewer 1 Report
The research is very important to study the priority of the community in field climate services. I note there is not enough information about what was the quantity and quality of the farmers’ capacity building training. We can use "The statistical bias test" it provides a simple assessment of how different the predicted outcomes may be for select groups in your data. If we made this test or other similar test we can avoid Bias in the outcomes of research by predetermined ideas, prejudice, or influence in a certain direction. I think the methodology can be improved and concentrate on the measure and indicators of changing the needs of farmers.

Reviewer 2 Report
The author are encouraged to additionally explain what kind of hydro-climatic information are meant and what kind of this information is the most important for analyzed areas.
L 176 Explain hydro climatic hazards in detail (what kind).
